# Design and Implementation of a Ku-Band High-Precision Blackbody Calibration Target

**DOI:** 10.3390/mi14010018

**Published:** 2022-12-21

**Authors:** Jie Liu, Zhenlin Sun, Guangmin Sun, Yu Li, Tong Cao, Wenjie Tang

**Affiliations:** Faculty of Information Technology, Beijing University of Technology, Beijing 100124, China

**Keywords:** blackbody calibration target, high emissivity, high temperature uniformity, finite element method

## Abstract

Microwave radiometers can be used in human tissue temperature measurement scenarios due to the advantages of non-destructive and non-contact temperature measurement. However, their accuracy often cannot meet the needs of practical applications. In this paper, a Ku-Band high-precision blackbody calibration target is designed to provide calibration for microwave radiometers and meet the requirements of a high temperature-measurement accuracy and high temperature-measurement resolution. From a practical application point of view, the blackbody calibration target needs to have the characteristics of high emissivity and high temperature uniformity. However, previous studies on blackbody calibration targets often focused on the scattering characteristics or temperature uniformity of the calibration target separately, and thus lack a comprehensive consideration of the two characteristics. In this paper, the electromagnetic scattering model and the temperature-distribution model of the calibration target are established through the multi-physical simulation combined with the Finite Element Method. Then, according to the simulation results of the two characteristic models, the structural parameters and composition of the coated cone array are continuously optimized. In addition, to achieve high-precision temperature control of the blackbody calibration target, this paper studies three PID controller parameter self-tuning algorithms, namely, BP-PID, PSO-PID and Fuzzy-PID for the optimal parameter tuning problem of traditional PID algorithms and determines the optimal temperature-control algorithm by comparing the performance of heating and cooling processes. Then, the blackbody calibration target is processed and manufactured. The arch test system is used to validate the reflectance of the calibration target, the emissivity is calculated indirectly, and the temperature-distribution uniformity of the temperature-control panel of the calibration target is tested by a multi-point distribution method. Finally, the uncertainty of the brightness temperature of the blackbody calibration target is analyzed.

## 1. Introduction

Due to the complexity of the internal structure of the human body and the influence of the external background environment, when a microwave radiometer is applied to measure the temperature of human tissue, the temperature measurement accuracy is often difficult to meet the requirements [1,2]. Therefore, it is necessary to provide calibration for the human body temperature microwave radiometer through a precise blackbody calibration target to improve the temperature measurement accuracy. In addition, according to the relationship between the frequency of a microwave and its penetration depth in the human body [3,4], only by determining the appropriate operating frequency band can the microwave radiometer correctly receive the radiation brightness temperature of the human tissue to be measured. This paper specifically measures the temperature of human multi-layer skin tissue and selects the Ku-Band as the working frequency band of the blackbody calibration target [5]. The design principle and design method of the precise blackbody calibration target in the Ku-Band are introduced below.

The blackbody calibration target is to convert the energy with a physical temperature of T into microwave energy with a brightness temperature of $T_{e}$ at an emissivity close to 1 and radiate it out. To achieve the best performance of the blackbody calibration target, it is necessary to: (1) optimize the electromagnetic characteristics of the blackbody calibration target to make the emissivity the highest [6]; and (2) analyze the physical temperature characteristics of the blackbody calibration target to ensure it has good temperature uniformity, stability and a small temperature gradient [7,8].

According to Kirchhoff’s radiation law [9,10,11], the sum of the reflectivity and emissivity of a microwave blackbody calibration target is 1. Designing a microwave blackbody calibration target with high emissivity means designing a structure with a high electromagnetic absorption rate and a low electromagnetic reflectivity [12,13]. Therefore, the analysis of electromagnetic scattering characteristics can provide important information for the design and measurement of a high-emissivity blackbody calibration target. There have been some studies which focused on the analysis and calculation of blackbody scattering characteristics. In recent years, Finite Difference Time Domain (FDTD) is often used for establishing an electromagnetic wave scattering model of blackbody [14,15,16]. Then, for an infinite cone array, the reflectivity can be solved based on the Radar Cross Section (RCS), while for a finite cone array, the reflectivity can be solved by integrating the full-space scattering power. Other studies have pointed out that the cone array is usually a periodic structure. The reflectivity of infinite periodic array structures can be evaluated more accurately using Floquet mode [17,18,19,20].

The temperature characteristics of the blackbody calibration target are independent of electromagnetic characteristics, and the most common analysis method is to use finite element software to conduct a thermal simulation analysis of the temperature field of the blackbody calibration target [21,22,23]. Based on the analysis, the factors affecting the uniformity of the temperature field can be discovered. The shape, material and heating conditions of the cavity structure can then be designed to give the blackbody calibration target a good temperature uniformity.

Although the internal structure of the blackbody calibration target has a great influence on the temperature uniformity, the control of its temperature is also a key point. The cavity of a blackbody radiation target needs to obtain a uniform temperature distribution, which requires a higher standard for temperature control. At present, the temperature control system in the black body calibration source mostly uses the traditional PID control algorithm, its parameters often need to be adjusted by experienced experts for many tests and debugging, the process is troublesome and time-consuming and laborious. Some researchers have combined the traditional PID control with fuzzy control [24,25], which can give rise to the full advantages of fuzzy control in dynamic control with good rapidity and make the stability of the system meet the requirements through the fine control of PID. With the in-depth study of intelligent algorithms and the good self-organization ability of neural networks, the research of PID algorithms based on neural networks in the field of temperature control has gradually increased [26,27]. Compared with traditional PID control, this method has significant advantages related to its adaptability, robustness and control quality.

The main work of this paper includes: (1) Due to the periodicity of the blackbody structure, this paper first uses the Floquet mode to study the scattering characteristics of an infinite array [28]. Then, according to the spatial symmetry, this paper proposes to use a two-dimensional scattering model to explore the two-dimensional pattern on the XOZ plane to further analyze the scattering characteristics of the finite array. The above methods greatly improve the computing efficiency and reduce the requirements for hardware devices. (2) The scattering properties and temperature uniformity of calibration targets are discussed separately in the existing literature. This paper notes that the shape and structure of the coated cone array will affect both its scattering characteristics and the temperature uniformity. Under the premise of satisfying both the emissivity and the temperature uniformity, a new design method is determined to find a set of design parameters to optimize the performance of the calibration target. (3) Several advanced PID algorithms are analyzed and compared, and an optimal algorithm is selected to be applied to the blackbody calibration target to obtain faster and more accurate temperature control. (4) We complete the processing and installation of the blackbody calibration target, test the emissivity and temperature uniformity of the equipment, and finally, introduce an uncertainty analysis.

The overall design process of the blackbody calibration target is shown in Figure 1.

## 2. Theory Analysis

### 2.1. Analysis of Electromagnetic Scattering Characteristics

The goal of the paper is to design a small blackbody calibration target, and therefore, the size of cone array is limited. However, in the specific analysis of the scattering characteristics of the cone array, it can be assumed that the size of the array is infinite. This paper then applies Floquet mode to analyze the surface scattering field. The surface scattering electric field distribution can be expressed as:(1)$$E^{s}=E_{P}^{s}\left(x,y,z\right)e^{-jk_{x}x}e^{-jk_{y}y}\;$$
where $k_{x}=-k\sin\theta_{i}\cos\varphi_{i}$, $k_{y}=-k\sin\theta_{i}sin\varphi_{i}$ are the propagation constants in the x and y directions, respectively, $\theta_{i}\;,\varphi_{i}$ represent the angle of the incident wave, k represents the wave number in free space, and $E_{P}^{s}\left(x,y,z\right)$ is a periodic function in the x and y directions, which can be expanded into Fourier series:(2)$$E_{P}^{s}\left(x,y,z\right)=\sum_{m=-\infty }^{\infty }\sum_{n=-\infty }^{\infty }E_{P_{mn}}^{s}\left(z\right)e^{-j\frac{2m\pi }{P_{x}}x}e^{-j\frac{2n\pi }{P_{y}}y}$$
where $P_{x}$ and $P_{y}$ are the periods of the array structure in the x and y directions. Substituting (2) into (1) produces:(3)$$E^{s}=\sum_{m=-\infty }^{\infty }\sum_{n=-\infty }^{\infty }E_{P_{mn}}^{s}\left(z\right)e^{-jk_{xm}x}e^{-jk_{yn}y}$$
where $k_{xm}=k_{x}+\frac{2m\pi }{P_{x}},\;k_{yn}=k_{y}+\frac{2n\pi }{P_{y}}$ are the propagation constants of the 𝑚th and 𝑛th space harmonics in the 𝑥 and y directions, respectively, and the propagation constants in the 𝑧 direction are:(4)$$k_{zmn}^{2}=k_{mn}^{2}-k_{xm}^{2}-k_{yn}^{2}$$

The scattering electric field $E^{s}$ is:(5)$$E^{s}=\sum_{m=-\infty }^{+\infty }\sum_{n=-\infty }^{+\infty }E_{mn}^{s}e^{-jk_{xm}x-jk_{yn}y-jk_{zmn}z}$$

The scattered magnetic field $H^{s}$ is:(6)$$H^{s}=\sum_{m=-\infty }^{+\infty }\sum_{n=-\infty }^{+\infty }H_{mn}^{s}e^{-jk_{xm}x-jk_{yn}y-jk_{zmn}z}$$

The scattered field at the array aperture can be regarded as the superposition of many uniform plane waves with different amplitudes and propagation directions. Some plane waves whose real part of the normal propagation constant is not zero will leave the array surface and radiate again. The scattering far field is composed of these plane waves. The waves whose normal propagation constant is imaginary can only propagate along the surface of the array and form surface waves. Assuming that the space is lossless, the condition of re-radiation is $k_{zmn}^{2}>0$ and can be obtained from Equation (4):(7)$${\left(-\;\sin\theta_{i}\cos\varphi_{i}+\frac{m\lambda }{P_{x}}\right)}^{2}+{\left(-\sin\theta_{i}sin\varphi_{i}+\frac{n\lambda }{P_{y}}\right)}^{2}<1$$

According to the Ku-band discussed in this paper, as well as the vertical normal incidence of a single station and the design structure of the calibration target, the values of m and n are both zero. The diagram of radiated harmonics is shown in Figure 2.

### 2.2. Analysis of Temperature-Distribution Characteristics

The thermal characteristics between the adjacent cones of the thermal calibration target, or between the cones and the outside environment, mainly include heat radiation and heat conduction. The two characteristics will have a coupling effect in the process of heat transfer, which has an important impact on the thermal characteristics.

Between the radiating surfaces of the absorber, Siegal and Howell extended the Stefan–Boltzmann law to a model with *N* heat radiating surfaces to obtain the energy balance equation [29], as shown in Equation (8):(8)$$\sum_{i=1}^{N}\left(\frac{\delta_{ji}}{\varepsilon_{i}}-F_{ji}\frac{1-\varepsilon_{i}}{\varepsilon_{i}}\right)\frac{1}{A_{i}}Q_{i}=\sum_{i=1}^{n}\left(\delta_{ji}-F_{ji}\right)\sigma T_{i}^{4}$$
where N is the number of radiating surfaces, $\delta_{ji}$ is the Kronecker function, $\varepsilon_{i}$ represents the effective emissivity of surface i, $F_{ji}$ represents the radiation angle coefficient, $A_{i}$ represents the area of surface i, $Q_{i}$ represents the energy loss of surface i, σ is the Stefan–Boltzman constant and $T_{i}$ is the absolute temperature of surface i.

For two models with mutual radiation interaction, Equation (4) can be simplified to Equation (5) to describe the heat transfer relationship between the i-plane and *j*-plane:(9)$$Q_{i}=\frac{\sigma (T_{i}^{4}-T_{j}^{4})}{\frac{1-\varepsilon_{i}}{A_{i}\varepsilon_{i}}+\frac{1}{A_{i}F_{ji}}+\frac{1-\varepsilon_{j}}{A_{j}\varepsilon_{j}}}$$

According to the total energy $q_{i}^{0}$ and effective radiation energy $q_{i}$ of each radiation surface, we can obtain Equation (10) and Equation (11):(10)$$\overset{N}{\underset{i=1}{\sum }}\left[\delta_{ji}-F_{ji}\left(1-\varepsilon_{i}\right)\right]q_{j}^{0}=\varepsilon_{i}\kappa \sigma T_{i}^{4}$$
(11)$$q_{i}=q_{i}^{0}-\sum_{j=1}^{N}F_{ji}q_{j}^{0}$$

For a radiation surface with a given temperature $T_{i}$, the effective radiation amount of each radiation surface can be deduced [17].

Heat conduction is the process of heat transfer between materials due to the temperature difference in the materials. Heat flows spontaneously from the high temperature material to the low temperature material through the object. Heat conduction is the main heat transfer mode of solids, and different substances have different thermal conductivity.

The heat conduction can be calculated according to Fourier’s law, and the calculation equation is as follows:(12)$$\left\{\begin{array}{c} q=-\lambda gradt\\ \phi =-\lambda Agradt \end{array}\right.$$
where 𝑞 is the heat flow density of heat conduction transfer, ϕ is heat conduction, and λ is the thermal conductivity, in which its size indicates the strength of the thermal conductivity of the object. Since heat energy is transmitted in the direction of a low temperature gradient, there is a minus sign in the equation.

## 3. Modeling and Simulation

### 3.1. Modeling of Electromagnetic Scattering Model

The calculation of the scattering properties of infinite periodic structures can be simplified using Floquet mode [30,31]. The paper uses a single cone as the computational unit, then applies periodic conditions for the boundaries of the single cone in the 𝑥 and 𝑦 directions, and similarly adds periodic ports above the cone. The reflection coefficient under the electromagnetic wave of a specific frequency can be obtained. As shown in Figure 3, the periodic boundary conditions in one direction only. However, the surface structure and materials of the cone model are relatively complex, making it easy to generate electromagnetic waves of different modes. Therefore, this design adds a perfect matching layer above the model to absorb electromagnetic waves of different models.

To verify the influence of the structural parameters of the calibration source on the scattering characteristics, it is also necessary to establish a three-dimensional model to analyze the scattering characteristics of the finite cone array. However, the computational complexity of the three-dimensional model is very high, and is difficult to complete with ordinary computers. When computing on a server with a 64-core CPU and 128-G memory, it can take up to one hour to obtain a set of three-dimensional patterns. However, for an effective conclusion, the current design requires studying the solution of hundreds of parameters. This paper investigates a method to reduce the computational complexity.

As shown in Figure 4, the pattern has strong spatial symmetry. What this design emphasizes is the change of the two-dimensional pattern on the XOZ plane with array parameters. A two-dimensional far-field scattering model is then proposed, which is shown in Figure 5. The method of building a two-dimensional model uses a certain cross section to study the scattering field. After verification, each group of results needs to be calculated for about 30 seconds on a PC platform with a 4-core CPU and an 8-G memory.

This paper uses the far-field node of Comsol software to calculate the radiation field at the specified location. The far-field node is used to calculate the scattering amplitude, so to determine the complex field at a particular location. The expression for the x component of the far-field electric field is:(13)$${\vec{E}}_{FFx}=E_{farx}\frac{e^{jkr}\hat{r}}{r}$$
where r is the radial distance in spherical coordinates, k is the wave vector of the medium and $E_{farx}$ is the scattering amplitude. It is worth noting that $E_{farx}$ is the magnitude of scattering in a particular direction, so it depends on angular position, but not radial position.

To verify this result, this paper simulates a perfect electric dipole and compares the simulation result with the analytical solution. The field distribution results of the dipole antenna can be divided into two terms, which are called the near field term ${\vec{E}}_{NF}$ and the far-field term ${\vec{E}}_{FF}$:(14)$$\vec{E}={\vec{E}}_{FF}+{\vec{E}}_{NF}$$
(15)$${\vec{E}}_{FF}=\frac{1}{4\pi \varepsilon_{0}}\left(\hat{r}\times \vec{p}\right)\times \frac{e^{-jkr}\hat{r}}{r}$$
(16)$${\vec{E}}_{NF}=\frac{1}{4\pi \varepsilon_{0}}\left[3r\left(\hat{r}\cdot \vec{p}\right)-\vec{p}\right]\left(\frac{1}{r^{3}}+\frac{jk}{r^{2}}\right)e^{-jkr}$$

The cone array cross section is regarded as a two-dimensional antenna model. Then, the background electric field is applied to the whole geometry field and the scattering amplitude of the PML is calculated as the scattering characteristics of the cone array. This assumes that the background field is set to $1\left[V/m\right]\cdot e^{jk\left(\hat{x}cos\theta +\hat{y}sin\theta \right)}$ to determine the direction of the background electrical field. In this model, the distance between the far-field node and the center of the cone array should meet the far-field condition. As shown in Figure 6, the relationship between the electric field of the dipole is calculated using the far-field node and the distance. By comparing the four solution methods of the far-field node, full-wave theory, near-field term and far-field term, it was found that the results of the far-field node and far-field theory coincide, and with the increase in the distance from the antenna, the results of the far-field node and full-wave theory tend to be consistent. This is because far-field nodes only consider 1/r radiation. Therefore, with the increase in distance, the contributions of $1/r^{2}$ and $1/r^{3}$ can be ignored, and the consistency with the full-wave theory will be improved. At the distance of three times the wavelength, the near-field radiation component is already very small, so the distance can be set to be nearly three times the wavelength in order to calculate the far-field scattering field.

Under the normal incidence of 15 GHz, the background electric field and relative electric field of the two-dimensional scattering model are shown in Figure 7, and the results are consistent with those of plane wave incidence. This shows that the model is correct.

### 3.2. Modeling of Temperature-Distribution Model

The coated cone is made of brass and installed on the bottom plate, and its gap is filled with thermal conductive silicone grease. To minimize heat loss, the coated cone array is surrounded by an outer shell and the bottom is covered with thermal insulation material. The corresponding diagram of the coating cone array is shown in Figure 8.

In Comsol software, the temperature-distribution model shown in Figure 9 can be established for a finite cone array. In this paper, the temperature range of the microwave radiometer temperature measurement device is between 303.15 K and 318.15 K. However, to fully explore the temperature distribution characteristics of the calibration targets, this paper simulates the calibration target with an operating temperature outside of the temperature measurement range. The temperature uniformity of the heating surface is ignored in the simulation, and the steady-state surface temperature is obtained at a set temperature of 333.15 K on the brass bottom surface and 293.15 K at the ambient temperature.

For the infinite periodic array, there is no heat dissipation to the surrounding area. Therefore, with the increase in the finite array size, the temperature characteristics of a single cone in an infinite model approach those of the most central cone in a finite array. However, the temperature condition of the cone at the boundary of a finite array cannot be simulated using the infinite model. Therefore, whether the model is finite or infinite in the electromagnetic simulation, the finite array model in Figure 9 is used to analyze the temperature-distribution characteristics.

As shown in Figure 9, the surrounding metal shell is considered ideally as a generalized heat flux condition and is set to have a thermal conductivity of 0.007 $W/m^{2}$. The top of the cone array is open and in direct contact with the environment, so the top is set as the natural convection model of the smoke pipe. The pipe diameter is set to a diameter of 110 mm and the pipe height is set to 100 mm to establish the cone array heat transfer model.

### 3.3. Simulation

The effects of the cone height, height to width ratio, coating material, coating thickness, top and bottom structure, number of cones, array arrangement and cone spacing on the array scattering characteristics and temperature uniformity are discussed, and the results in Table 1 are obtained.

Based on the above analysis of calibration source modeling, the second-generation blackbody calibration source model was established and the index optimization was completed after the structural parameters of the first-generation blackbody calibration source model were improved. Experiments were performed to compare the simulation results of the first-generation blackbody calibration target and the second-generation blackbody calibration target. The results are shown in Figure 10 and Figure 11. In Figure 11, a yellow box labeled “min” is used to indicate the minimum temperature of the cone array surface.

As shown in Figure 10, that the average emissivity of the second-generation calibration target is higher than that of the first-generation calibration target. At the same time, the emissivity of the second-generation blackbody calibration target is higher than 0.9999 in the band. As shown in Figure 9, the area proportion of the cone surface temperature difference less than 0.3 K in the second generation is roughly 91%, which is about 1.8% higher than that of the first generation. The maximum temperature difference of the cone surface of the first generation is 3.459 °C, while that of the second generation is 2.831 °C. This indicates that the second-generation calibration target is superior, which verifies the effectiveness of the optimization method.

## 4. Research of Temperature-Control Algorithms

In this section, the blackbody calibration target is taken as the control object, and the temperature-control algorithms include PID, PSO-PID, fuzzy-PID and BP-PID.

The temperature-control system of the blackbody calibration target is approximately equivalent to the first-order inertial element plus the delay element. The general mathematical form of the transfer function is:(17)$$G\left(s\right)=\frac{K}{Ts+1}e^{-\tau s}\;$$

This paper divides the temperature change situations into two types: heating and cooling. The transfer functions in the two cases are set as Equation (18) and Equation (19):(18)$$G_{1}\left(s\right)=\frac{T_{1}\left(s\right)}{U\left(s\right)}=\frac{114.03}{286.89s+1}e^{-9s}\;$$
(19)$$G_{2}\left(s\right)=\frac{T_{2}\left(s\right)}{U\left(s\right)}=\frac{-13.466}{206.96s+1}e^{-9s}\;$$

For the BP-PID algorithm, to optimize the parameters of the PID algorithm using the BP neural network, the number of hidden layers of the BP network, the learning rate of the weights, and the momentum factor can be adjusted. The learning rate has the greatest impact on the temperature-control accuracy, and the momentum factor will affect whether the temperature-control process produces oscillations. To prevent the output result from failing to converge, it is necessary to control the size of PID control increment Uk obtained after each training of the BP neural network and make the temperature control result converge to the set temperature value. In addition, for the transfer function of cooling, it is necessary to set the control rate of the PID parameter as a negative number to achieve the cooling effect. The experimental results show that when the hidden layer is 8, the learning rate is 0.0005, and the momentum factor is 0.3. Thus, an ideal heating effect can be obtained. However, it cannot be guaranteed that the cooling process can achieve an ideal effect every time. Since the weight matrix of the BP neural network is randomly assigned each time, it is easy for the BP network to fall into a local minimum.

The PSO-PID algorithm is based on the traditional PID algorithm in addition to the objective function of the particle swarm optimization algorithm and the objective loss function with time weight, and sets the three parameters P, I, D as the three position variables Kp, Ki and Kd. The number of iterations of particle swarm optimization are set to 10, the size of particle swarm optimization to 100 and the inertia factor to 0.6. Then, the PID optimization range is set. With the increase in the number of iterations, the particle swarm optimization algorithm dynamically changes the PID parameters so that the fitness of the objective function gradually decreases, reaching the effect of convergence to the set temperature value.

For the Fuzzy-PID algorithm, the number of output discrete samples in the Fuzzy controller is adjusted. The larger the value is, the faster the PID converges, although there will be a marginal decline effect. Compared with traditional PID regulation, Fuzzy-PID regulation is more intelligent, gradually refined and has a faster response speed and less overshoot.

Through the above analysis of the parameter tuning methods of three PID optimization algorithms, the effects of four temperature-control algorithm models are compared. The overall algorithm structure block diagram is shown in Figure 12.

To simulate the heating condition, this paper sets the target temperature as 30 °C and the initial temperature as 0°C. The heating curve is shown in Figure 13. For another cooling condition, this paper sets the target temperature as −30 °C and the initial temperature as 0 °C. The cooling curve is shown in Figure 14.

As shown in Figure 13, BP-PID has the best performance due to its small overshoot and the fastest response speed. Compared with traditional PID, PSO-PID and Fuzzy-PID algorithm have some improvements, but there is still a certain gap compared to the BP-PID algorithm.

As shown in Figure 14, the comparison of the above four algorithms that Fuzzy-PID has the largest overshoot, PSO-PID algorithm has small overshoot but the convergence speed is very slow, while BP-PID has the fastest convergence and small overshoot.

Considering the overshoot, convergence time, convergence accuracy and other indicators of the heating and cooling process, the final blackbody calibration source uses the BP-PID temperature-control algorithm. However, in the process of multiple tests, the temperature-control effect of BP-PID may be unstable. Therefore, when using this algorithm, the parameters of the model should be set reasonably to obtain the best effect.

## 5. Processing and Testing

### 5.1. Processing of Blackbody Calibration Target

To ensure that the above performance indicators of the calibration source simulation can be achieved, the engineering prototype of the calibration source needs to be precision machined. The main components of the prototype are described according to the processing process of the blackbody calibration source. As shown in Figure 15a, the brass matrix forms the cone array. Figure 15b shows the coated cone, whose surface is made of domestic P-0001 microwave-absorbing material with a density of 4.2 g/cm^3^ with an attenuation of −35 dB/cm at 10 GHz frequency. During coating, the coating and curing agent is mixed at a ratio of 100:3.5, and then the cone surface is painted and cured several times to reach the required coating thickness. Next, the coated cone is placed in an oven at 120 °C for 20 min to achieve full curing. Figure 15c coated cone array. The coated cone is fixed on the bottom plate of the thermal insulation chamber with M4 screws, and the array layout is designed according to the simulation. Figure 15d shows the temperature controller AI-516PD2G. The internal temperature-control algorithm is implemented using the BP-PID algorithm. The relevant parameters of the algorithm are set through the serial port command protocol. The measurement range is 0–1300 °C, the resolution is 0.1 °C, the maximum allowable error is ±3.9 °C and the accuracy class is 0.3 °C. Figure 15e shows the calibration source cavity structure. The outer side of the cavity is surrounded by thermal insulation materials. The shell is made of 6061 aluminum alloy. The entire cavity is fixed on the temperature-equalizing plate of the thermostatic heating table with screws, so as to realize the temperature control of the cone array coated in the cavity. Figure 15f shows the complete blackbody calibration source.

### 5.2. Electromagnetic Emissivity Test Scenario and Results

The practical testing scene includes the arched test system, vector network analyzer, 1–18 GHz double-ridged horn antenna and test cable. The diagram of the arched test system is shown in Figure 16, and the practical testing scene is shown in Figure 17.

The basic principle of the arch method is the dual antenna power method. A pair of antennas with the same frequency and polarization direction are used as the transmitting antenna and the receiving antenna, respectively. The transmitting antenna and the receiving antenna are placed on the same side of the tested sample. The maximum gain direction always points to the tested sample. The transmitting antenna generates excitation to the tested sample, and its reflected signal is received by the receiving antenna. The reflectivity of the tested sample is:(20)$$r=Pr+Gt+Gr-Pt-32.4-20lg\left(f\right)-20lg\left(L\right)$$
where *Pt*, *Pr* are the transmitting power of the transmitting antenna and receiving power of the receiving antenna, *Gt*, *Gr* are the gain of the transmitting antenna and the gain of the receiving antenna, *F* represents the test frequency and *L* represents distance from the phase center of the transmitting antenna to the phase center of the receiving antenna through the reflection path of the material.

The arch test system is used to locate the transmitting and receiving antennas, so that the maximum gain direction of the transmitting and receiving antennas always points to the tested sample. In addition, the absorbing material is laid on the arch frame ground. The steps to follow when using the arch test system for testing include: (1) Select the antenna with the same frequency (one transmitting antenna and one receiving antenna), and connect the antenna with the port of the vector network analyzer using high-performance test cables; (2) Install the antenna on the arch frame to ensure that the two antennas are aligned, and raise them to the top of the arch frame; (3) Place the pyramidal absorbing material on the test bench; (4) Set the frequency band of the network analyzer. Then select the S11 parameters for testing and reset the test parameters to zero; (5) Place the tested blackbody calibration on the metal plate and record the parameters.

Reflectivity and emissivity are tested and the results are shown in Table 2. It can be seen that, in the frequency band, the emissivity e of the blackbody calibration target is greater than 0.998.

### 5.3. Temperature Distribution Test Scenarios and Results

At the ambient temperature of 24 °C, an electronic thermometer, K-type thermocouple, thermal conductive silicone grease and other materials were used to test the thermostatic heating table. The device was set at 100 °C, and the target temperature was reached in 3 min and 25 s. The test restarted after 5 min of stabilization. The minimum measured data was 97 °C and the maximum was 102 °C at the initial time. The device was then placed at a constant temperature for 30 min and tested again, with a minimum temperature of 99 °C and a maximum of 101 °C. The temperature test points of the thermostatic heating table are shown in Figure 18, and the specific data are shown in Table 3.

### 5.4. Uncertainty Analysis

The main sources of the uncertainty of the brightness temperature of the blackbody calibration target are: the uncertainty introduced by the temperature measurement of the blackbody calibration target $u_{1}$; the uncertainty introduced by the calculation of the effective emissivity of the blackbody cavity at variable temperature $u_{2}$; and the uncertainty introduced by the ambient temperature $u_{3}$.

The uncertainty of the temperature measurement of the blackbody calibration target includes: the uncertainty introduced by the calibration of temperature sensor $u_{11}$; the uncertainty introduced by the stability of temperature sensor $u_{12}$; the uncertainty introduced by the electric measuring instrument $u_{13}$; the uncertainty of heat conduction in the cavity bottom $u_{14}$; the uncertainty introduced by temperature control stability $u_{15};$ and the uncertainty introduced by the uniformity of the cavity bottom temperature $u_{16}$. Among them, $u_{15}$ is mainly determined by the BP-PID temperature-control algorithm used in this paper.

The uncertainty $u_{1}$ introduced by the temperature measurement of the blackbody calibration target can be calculated by the following equation:(21)$$u_{1}=\sqrt{u_{11}^{2}+u_{12}^{2}+u_{13}^{2}+u_{14}^{2}+u_{15}^{2}+u_{16}^{2}}\;$$

The combined standard uncertainty is:(22)$$u_{c}=\sqrt{u_{1}^{2}+u_{2}^{2}+u_{3}^{2}}$$

The temperature sensor used in this paper is a K-type thermocouple, and the uncertainty $u_{11}$ introduced by its calibration is 0.1 K. According to the test, the long-term stability of the K-type thermocouple is 1.3 K, and the introduced uncertainty $u_{12}=1.3/\sqrt{3}\approx 0.75$ K.

The electrical measuring instrument used in this paper is AI-516PD2G, and the uncertainty $u_{13}$ introduced by it is 0.3 K. The uncertainty $u_{14}$ obtained through the simulation is 0.3 K, because the proportion of the area with a temperature difference between the cone surface and the bottom that is less than 0.3K exceeds 90%. The uncertainty $u_{15}$ is very small and can be ignored. The uncertainty $u_{16}$ can be obtained from the multi-point distribution experiment and has a value of 0.44 K.

The effective emissivity of the blackbody calibration target is 0.999±0.0008, and therefore the calculation result of $u_{2}$ is 0.0008$/\sqrt{3}$. Similarly, the influence of ambient temperature on brightness temperature is small enough that it can be ignored.

Finally, through calculation, we achieve the combined standard uncertainty $u_{c}\;$= 0.9729.

## 6. Conclusions and Outlooks

The paper describes the basic principle of the calibration of microwave temperature measurement system, discusses the theoretical basis of the blackbody calibration target design, and summarizes the method of designing a calibration target based on electromagnetics simulation. Then, the scattering characteristics and temperature-distribution characteristics of the blackbody calibration target are analyzed theoretically. Based on the finite element method in the simulation software Comsol, the corresponding scattering model and heat transfer model are established. Combined with the simulation results of the two models, the parameter optimization results of the blackbody calibration source are given. Through comparative experiments, it is proven that the method can effectively improve the blackbody calibration source in terms of its emissivity and temperature uniformity. To achieve faster and more accurate temperature control, this paper compares three advanced PID algorithms—BP-PID, PSO-PID, and Fuzzy PID—by simulation. BP-PID has the advantages of a small overshoot and fast response in both heating and cooling conditions. Therefore, this algorithm is selected as the temperature-control algorithm of the blackbody calibration source. After completing the above design, this paper realized the processing of blackbody calibration source, and completed the emissivity test based on the arch test system. At the same time, the multi-point distribution method is used to simulate the temperature uniformity. The test results showed that the emissivity of the designed blackbody calibration target is higher than 0.998 in Ku-band, and the temperature uniformity is good.

In conclusions, the design of the Ku-Band blackbody calibration target has been completed. However, there are still some deficiencies that need to be addressed.

The blackbody calibration target prototype should be further tested, and the influence of emissivity and temperature-distribution uniformity on the overall calibration accuracy of the microwave radiometer should be given.The heat transfer model is established on the premise that the temperature of the bottom surface of the cone array is consistent and uniform, and there will be temperature differences on the plane in practice. Moreover, the effect of air convection is not considered. Later, the definition of air convection will be added to the modeling, and the temperature-distribution gradient of the bottom surface of the calibration target will be set.

## Figures and Tables

**Figure 1 micromachines-14-00018-f001:**
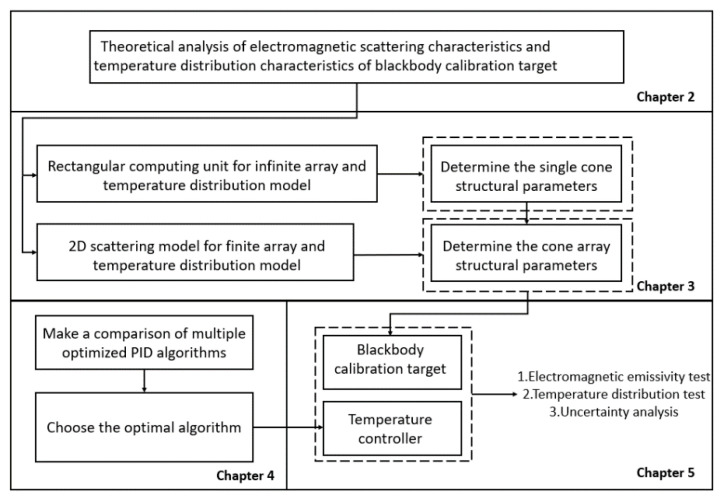
The overall design of this paper.

**Figure 2 micromachines-14-00018-f002:**
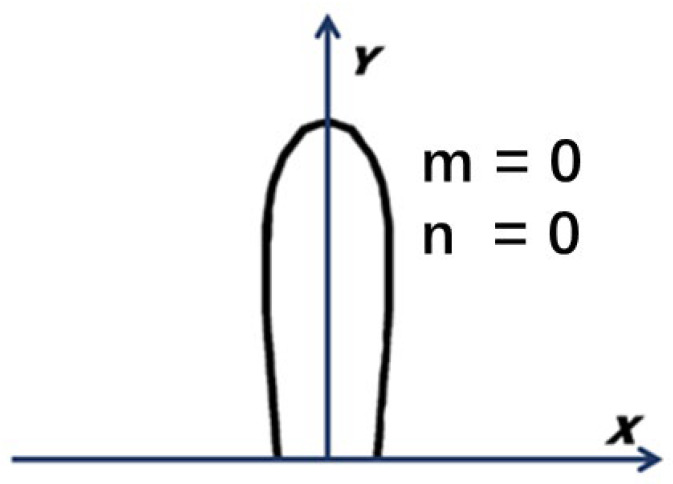
The diagram of radiated harmonics.

**Figure 3 micromachines-14-00018-f003:**
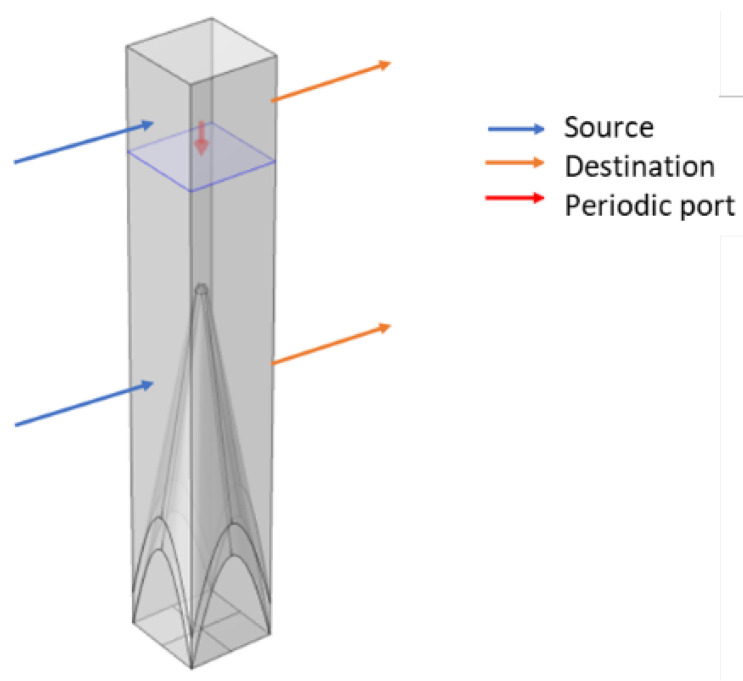
Rectangular computing unit model.

**Figure 4 micromachines-14-00018-f004:**
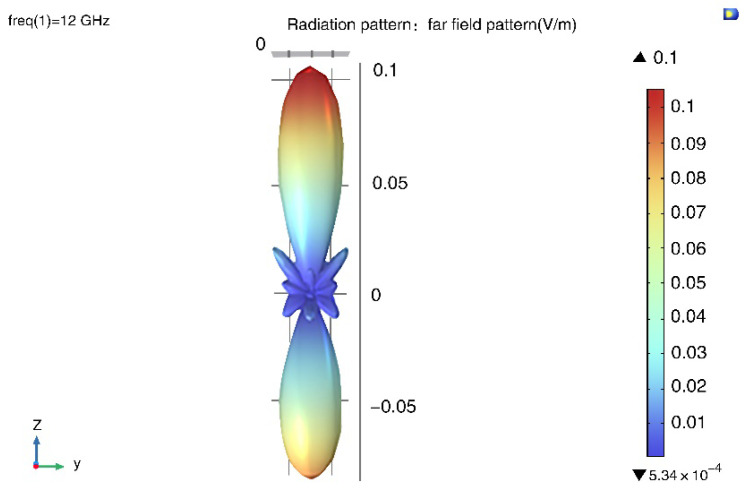
Three-dimensional radiation pattern.

**Figure 5 micromachines-14-00018-f005:**
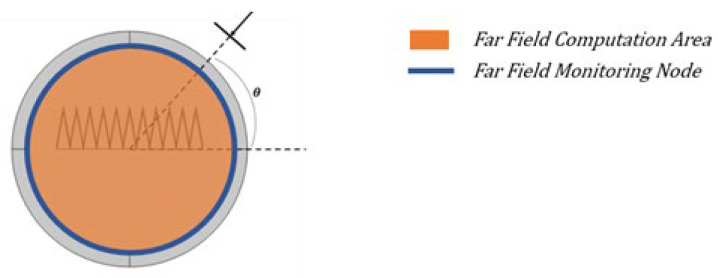
2D Scattering model.

**Figure 6 micromachines-14-00018-f006:**
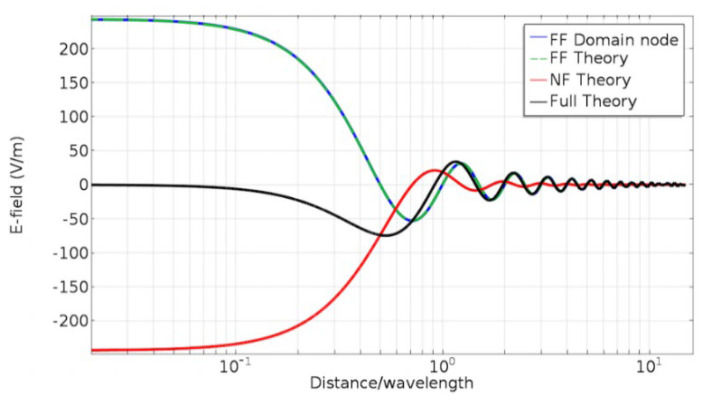
The relationship between the electric field of the dipole $\vec{p}$ and the distance.

**Figure 7 micromachines-14-00018-f007:**
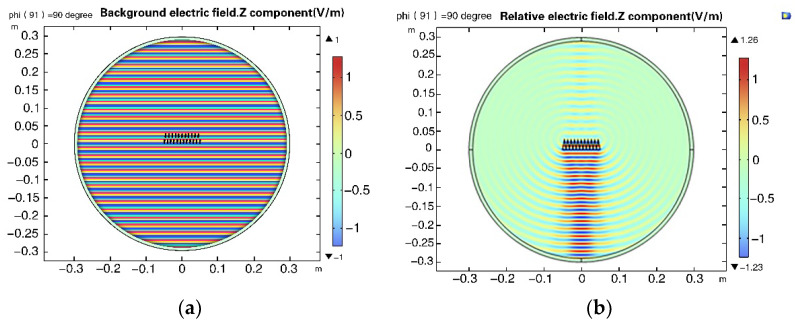
Electric field simulation results of the 2D scattering model. (**a**) Background electric field. (**b**) Relative electric field.

**Figure 8 micromachines-14-00018-f008:**
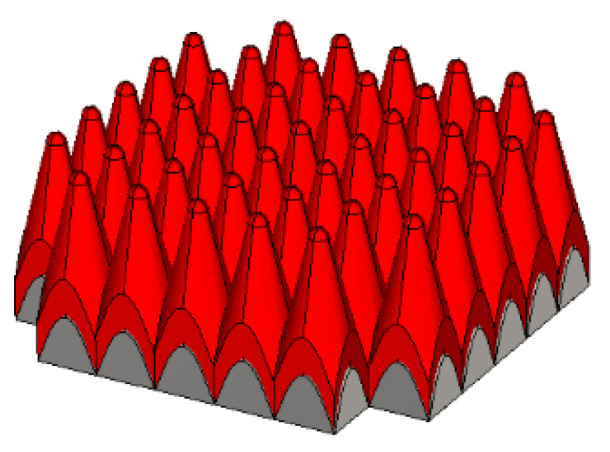
The diagram of the coated cone array.

**Figure 9 micromachines-14-00018-f009:**
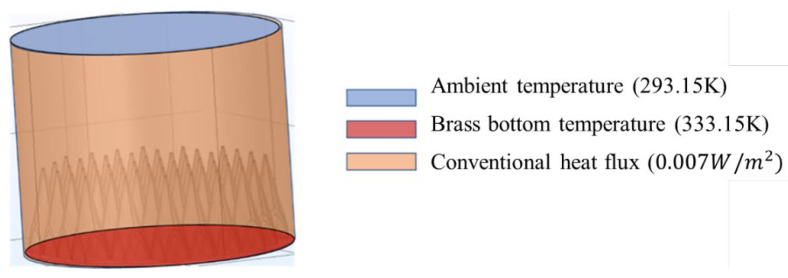
Temperature-distribution model of the coated cone array.

**Figure 10 micromachines-14-00018-f010:**
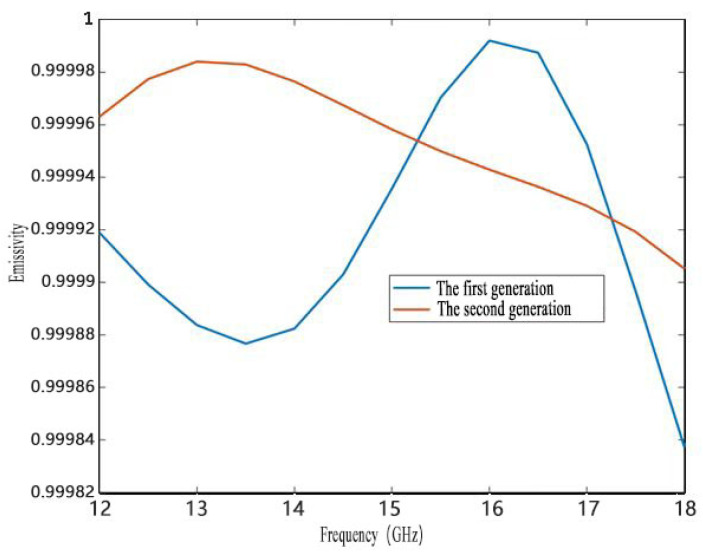
Emissivity results of two generations of calibration targets.

**Figure 11 micromachines-14-00018-f011:**
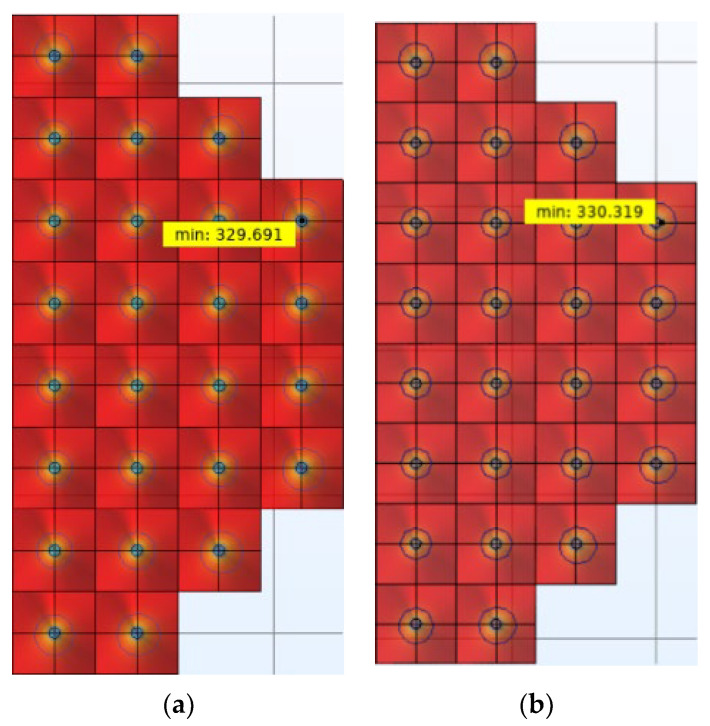
Temperature uniformity results of two generations of calibration targets. (**a**) The first-generation calibration target. (**b**) The second-generation calibration target.

**Figure 12 micromachines-14-00018-f012:**
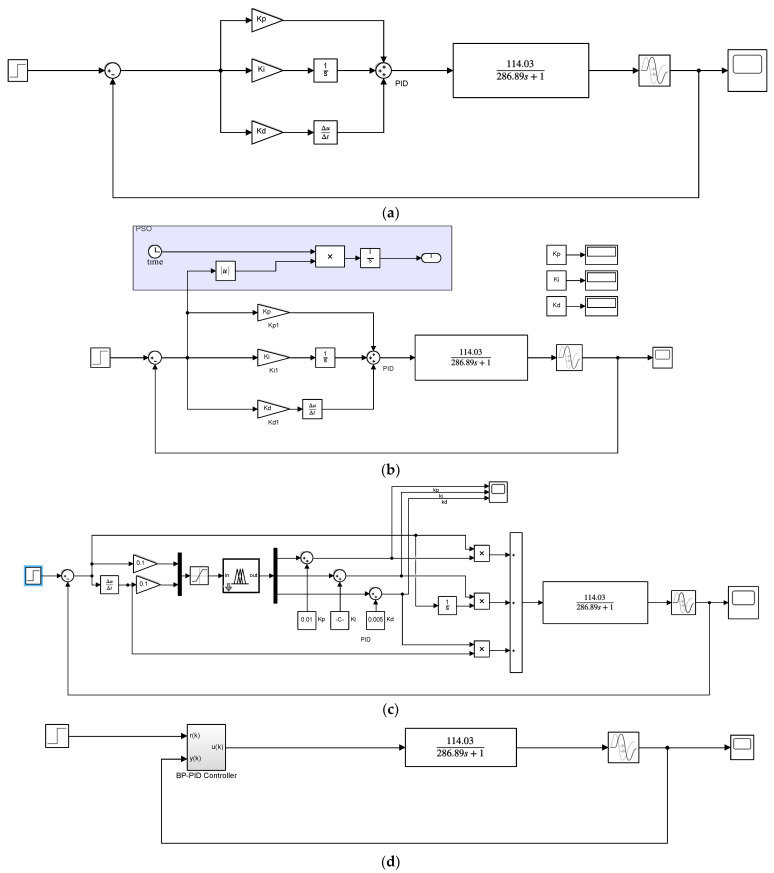
Block diagram of four temperature-control algorithms. (**a**) PID; (**b**) PSO-PID; (**c**) Fuzzy-PID; (**d**) BP-PID.

**Figure 13 micromachines-14-00018-f013:**
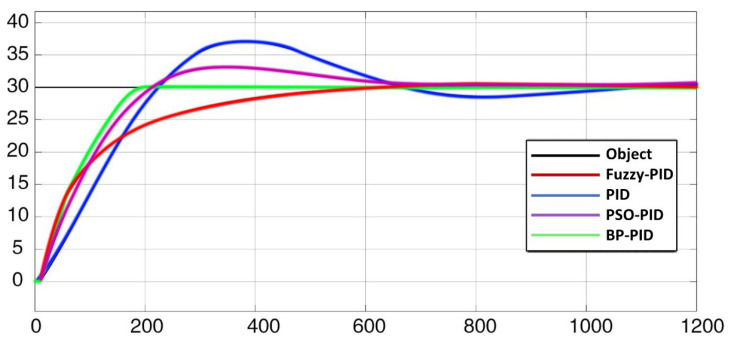
Heating response curve of the controlled object.

**Figure 14 micromachines-14-00018-f014:**
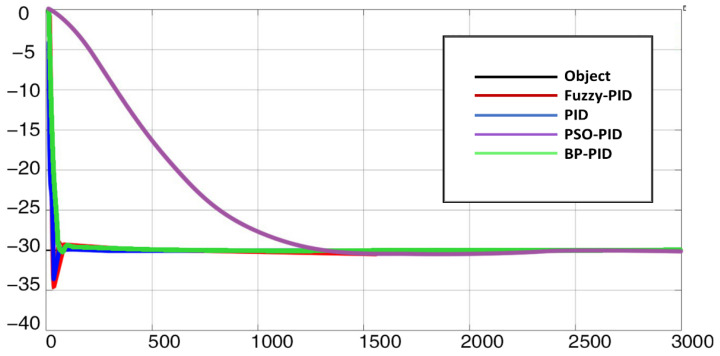
Cooling response curve of the controlled object.

**Figure 15 micromachines-14-00018-f015:**
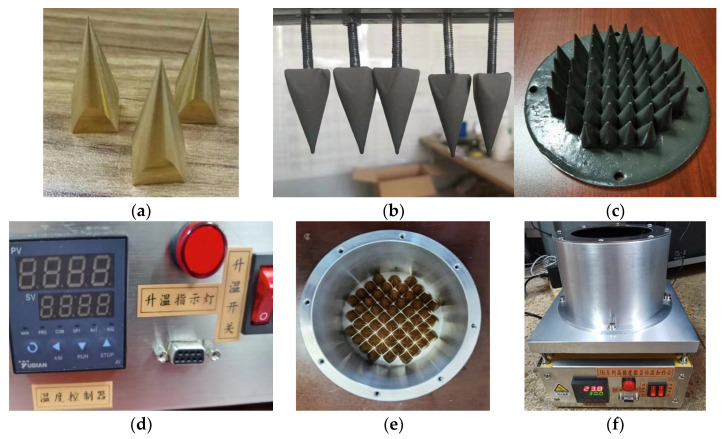
The figures for the blackbody calibration target. (**a**) Metal cone; (**b**) Coated cone; (**c**) Coated cone array; (**d**) Temperature controller; (**e**) The cavity of the calibration target; (**f**) Complete blackbody calibration target.

**Figure 16 micromachines-14-00018-f016:**
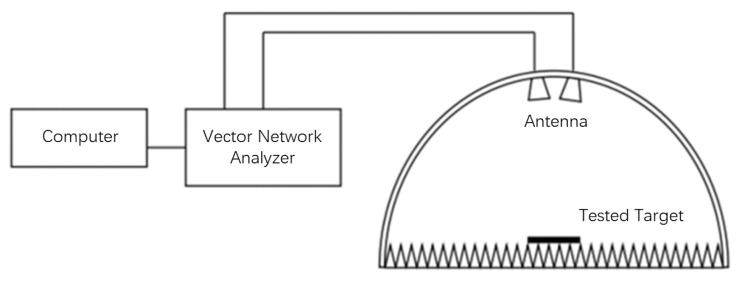
The diagram of the arched test system.

**Figure 17 micromachines-14-00018-f017:**
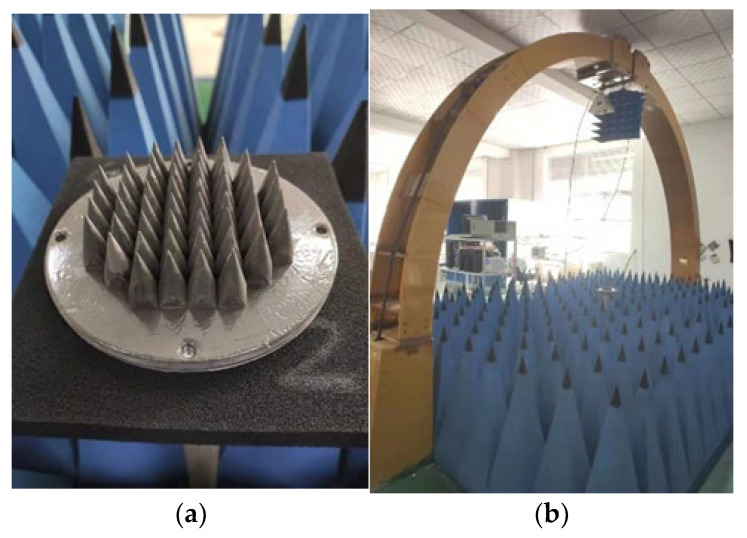
The practical testing scene. (**a**) Tested target; (**b**) Arch test system.

**Figure 18 micromachines-14-00018-f018:**
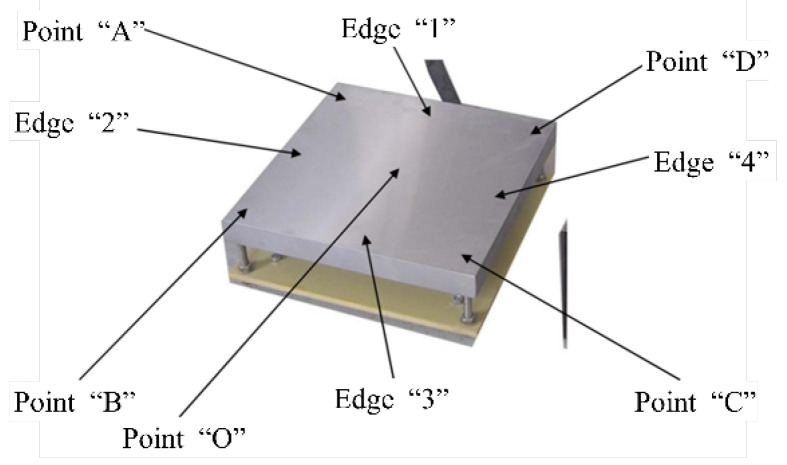
Temperature test points.

**Table 1 micromachines-14-00018-t001:** Structure parameters of two generations of blackbody calibration source prototypes.

Parameter	The First Generation	The Second Generation
Height	36 mm	39 mm
Height to width ratio	3	3.5
Coating material	CR112	CR117
Coating thickness	0.8 mm	0.7 mm
Cone top shape	Sphere	Flat
Cone bottom shape	Square	Square
Number of cones	52	52
Arrangement	Octagonal	Octagonal
Distance between cones	0	1 mm

**Table 2 micromachines-14-00018-t002:** The electromagnetic test results of the blackbody calibration target.

Frequency (GHz)	12	13	14	15	16	17	18
Reflectivity	0.0018	0.001	0.0009	0.0013	0.0004	0.0002	0.0014
Emissivity	0.9982	0.999	0.9991	0.9987	0.9996	0.9998	0.9986

**Table 3 micromachines-14-00018-t003:** The results of the temperature test.

Time (min)	Set Temperature (°C)	Actual Results of Surface Temperature (°C)								
		O	A	B	C	D	1	2	3	4
10	100	102	97	99	99	97	97	98	97	98
30	100	101	98	100	100	98	98	100	99	100

## Data Availability

Not applicable.

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
