# Peer review of "Design and Implementation of a Ku-Band High-Precision Blackbody Calibration Target"

_micromachines, 2022, doi:10.3390/mi14010018_

Round 1

Reviewer 1 Report (Previous Reviewer 1)

The manuscript has been greatly improved after revision. There are some problems that need to be modified.

1.          Although my initial thought is to recommend that you use constant power boundary conditions, which are more in line with the actual condition of using film heaters. But your method also seems feasible, and the aim is to study the temperature difference from the metal platform to the radiant cone. It can be highlighted in the paper. In addition, the difference between the cone and the metal platform should be considered in the evaluation of temperature performance, since your temperature sensor is measuring the temperature of the metal platform and not the actual radiant surface. The temperature values can be measured by reference to simulation results or by radiation thermometers.

2.          Please give better quality Figures, some of the text in the images is difficult to read.

3.          In Fig. 18, please directly give the wavelength emissivity results, which is more intuitive.

4.          Please give the values, results and the temperature range or temperature point assessed of the 5.4 Uncertainty Analysis. In addition, u11 contains u12.

5.          Please put a space before the unit.

Author Response

Dear reviewer

Regards

Liu Jie

Reviewer 2 Report (Previous Reviewer 2)

Authors said that they will address t the effects of the variability of emissivity and temperature distribution uniformity on the overall accuracy of radiometer performance in a future work. Let's hope it is true.

Author Response

Dear reviewer

Regards

Liu Jie

This manuscript is a resubmission of an earlier submission. The following is a list of the peer review reports and author responses from that submission.

Round 1

Reviewer 1 Report

This article describes the design and implementation of ku-band high-precision blackbody calibration target. It seems more like an engineering report than a scientific study, And the results are not impressive. Here are some problem:

1)        Please explain the innovations and scientific problems solved in this paper.

2)        Is the the electromagnetic scattering model and the temperature distribution model first proposed by you? If not, what contributions have you made?

3)        Lines 38-39, what does “the physical temperature distribution on the surface of calibration target is relatively concentrated” mean? How to ensure the measurement accuracy?

4)        Lines 118-138, only surface-surface radiation seems to have been considered, but solid and convective heat transfer are also important considerations for your size.

5)        Figure 4. What does “Temperature constraint” mean.

6)        lines 200-201, you seem to have set the brass bottom surface to 333.15 K, which is not accurate

7)        Figure 6. I can hardly see the specific temperature field distribution under your color scale.

8)        The difference between simulation results and experimental results of emissivity and temperature field uniformity is too large. Please check whether your model is correct.

9)        Is the simulation using your proposed model or COMSOL built-in model? Please indicate your contribution.

10)     Is the temperature control algorithm used in this work your own or Yudian PID's own? Please indicate your contribution.

Author Response

Dear reviewer, thank you for your comments.

Kind regards.

Reviewer 2 Report

General comment:

The authors described a blackbody calibration target design, where the scattering characteristics and temperature distribution of the blackbody are analyzed theoretically through simulations.

Then the blackbody calibration target is realized and tested.

The work is interesting, but some improvements are necessary in the manuscript structure and content.

Furthermore, the language needs a revision.carefully

Specific comments:

- Introduction: the state of the art of black-body calibration target is missing, both for space-based and ground-based radiometer. It is not clear what is the application of the proposed blackbody.

- Why the research frequency is the Ku-band? Which kind of applications are expected for the sensors with that blackbody target?

-Lines 115-117: the scientific meaning of the sentence is not clear

-In eq. 6 and 7, some variables are not defined.

-Before section 3, the description and scheme of the proposed calibration target is necessary (like fig.12).

-Line 190-193: without an explicative picture, it is hard to understand

-figure 6: the unit of measurements is missing. In the yellow box, “min” refers to the minimum temperature? Please, explain in the caption

-Fig. 7: how is the emissivity computed?

-Overall, after the test measurements on the manufactured blackbody, the authors should explain what are the effects of the variability of emissivity and temperature distribution uniformity on the overall accuracy of a radiometer calibration.

Author Response

(The authors gave the same response as above.)

Round 2

Reviewer 1 Report

The error in the temperature field simulated are understandable. This is mainly caused by the difference of contact heat conduction the between the ideal condition and the actual structural plane. But in this study, the constant temperature boundary conditions you used were inaccurate and unrealistic. It's going to bring in an infinite reservoir to keep the temperature constant. Make your simulation result more uniform than the actual. In addition, your temperature sensor appears to be in the metal platform below, which does not correspond to your simulation results, nor can accurately obtain the temperature of cone radiant surface.

In the simulation of microwave optics, the reason for such a large difference is not clear to me, but in my experience optical simulations should rarely show such a large difference. I recommend that you use more specialized optical simulation software and models.

Is the two-dimensional far-field scattering calculation model proposed for the first? If so, describe it in detail and computational costs using the 3D actual and 2D periodic physics models. It seems interesting.

Finally, the lack of description of the coating absorption parameters and the accuracy of the temperature measurement system in the experimental section is significant.

In general, I think this work uses the existing simulation methods in theory, and lacks innovation, adequate reflection of research results, and no outstanding results in engineering.

Reviewer 2 Report

Minor Comments:

Fig. 9: In the yellow box, “min” refers to the minimum temperature? Please, explain in the caption

-Line 396: the equation number is 17, not 21

-Authors have not discussed in tha manuscript the effects of the variability of emissivity and temperature distribution uniformity on the overall accuracy of radiometer performance (trying a quantitative estimation of uncertainties described in the author comments)
